# Pulmonary Aspergillosis in Immunocompromised Critically Ill Patients: Prevalence, Risk Factors, Clinical Features and Diagnosis—A Narrative Review

**DOI:** 10.3390/jof11090617

**Published:** 2025-08-24

**Authors:** Maria Grazia Bocci, Laura Cascarano, Giulia Capecchi, Antonio Lesci, Valerio Sabatini, Dorotea Rubino, Giulia Valeria Stazi, Gabriele Garotto, Stefania Carrara, Antonella Vulcano, Chiara Gori, Franca Del Nonno, Daniele Colombo, Laura Falasca, Emanuela Caraffa, Stefania Cicalini, Carla Fontana

**Affiliations:** 1Clinical and Research Department, National Institute for Infectious Diseases “Lazzaro Spallanzani”—IRCCS, 00149 Rome, Italy; mariagrazia.bocci@inmi.it (M.G.B.); valerio.sabatini@inmi.it (V.S.); dorotea.rubino@inmi.it (D.R.); giuliavaleria.stazi@inmi.it (G.V.S.); gabriele.garotto@inmi.it (G.G.); 2Anesthesia and Intensive Care, San Camillo-Forlanini Hospital, 00152 Rome, Italy; lcascarano@scamilloforlanini.rm.it (L.C.); cgori@scamilloforlanini.rm.it (C.G.); 3Department of Emergency, Intensive Care Medicine and Anesthesia, Fondazione Policlinico Universitario A. Gemelli IRCCS, 00168 Rome, Italy; antonio.lesci01@icatt.it; 4Istituto di Anestesiologia e Rianimazione, Università Cattolica del Sacro Cuore, 00168 Rome, Italy; 5Laboratory of Microbiology and Biological Bank Unit, National Institute for Infectious Diseases “Lazzaro Spallanzani”—IRCCS, 00149 Rome, Italy; stefania.carrara@inmi.it (S.C.); antonella.vulcano@inmi.it (A.V.); carla.fontana@inmi.it (C.F.); 6Pathology Unit, National Institute for Infectious Diseases “Lazzaro Spallanzani”—IRCCS, 00149 Roma, Italy; franca.delnonno@inmi.it (F.D.N.); daniele.colombo@inmi.it (D.C.); 7Laboratory of Electron Microscopy, National Institute for Infectious Diseases “Lazzaro Spallanzani”—IRCCS, 00149 Rome, Italy; laura.falasca@inmi.it; 8Systemic and Immune Depression-Associated Infections Unit, Clinical Department, National Institute for Infectious Diseases “Lazzaro Spallanzani”—IRCCS, 00149 Roma, Italy; emanuela.caraffa@inmi.it (E.C.); stefania.cicalini@inmi.it (S.C.)

**Keywords:** *Aspergillus*, pulmonary aspergillosis, invasive aspergillosis, immunocompromised patients, antifungal treatment

## Abstract

Aspergillosis in immunocompromised individuals is a serious and potentially life-threatening infection, as the weakened immune system cannot effectively fight the *Aspergillus* fungus. This review provides an in-depth examination of aspergillosis in patients with various conditions that compromise immunity, including hematological disorders, HIV, SARS-CoV-2 pneumonia, influenza, and those who have undergone solid organ transplants. The clinical manifestations of aspergillosis are influenced by factors such as the host’s underlying comorbidities, immune response, and immune suppression due to medications or treatments. The review delves into the epidemiology of aspergillosis, exploring various risk factors that predispose individuals to infection. It also discusses the wide range of clinical symptoms, highlighting the challenges in diagnosis and the importance of early detection. The review contrasts traditional diagnostic approaches with emerging molecular techniques, emphasizing the role of advanced diagnostics in improving outcomes. A proposed clinical decision-making flowchart is provided to assist healthcare professionals in managing suspected cases of aspergillosis. In addition to diagnostic challenges, the review addresses antifungal prophylaxis, pre-emptive therapy, and the growing concern of pharmacological resistance to antifungal agents. It concludes with a discussion of future research directions, underscoring the need for improved therapeutic strategies and preventative measures in immunocompromised patients to reduce the burden of this severe fungal infection.

## 1. Introduction

Aspergillosis is an opportunistic fungal infection caused by molds of the genus *Aspergillus*, characterized by a wide spectrum of clinical manifestations. These range from allergic forms, such as allergic bronchopulmonary aspergillosis (ABPA) and severe asthma with fungal sensitization (SAFS), to progressive chronic forms like chronic pulmonary aspergillosis (CPA), and finally to acute, potentially lethal forms of invasive aspergillosis (IA) [1] (Figure 1). In recent years, novel clinical entities have emerged in association with severe viral infections, including influenza-associated pulmonary aspergillosis (IAPA) and COVID-19—associated pulmonary aspergillosis (CAPA), which have been described in critically ill patients admitted to intensive care units, even in the absence of classical immunosuppressive risk factors [2,3]. From an etiological standpoint, *Aspergillus fumigatus* remains the principal pathogen responsible for aspergillosis [1,4]. Other relevant species include *Aspergillus flavus*, *Aspergillus niger*, *Aspergillus terreus*, *Aspergillus nidulans*, and *Aspergillus calidoustus*, which are among the most frequently involved in human disease [1]. These names actually refer to complexes of cryptic species that are morphologically similar [1]. Although *A. fumigatus* sensu stricto is by far the most prevalent, cryptic related species (e.g., *Aspergillus lentulus*, *Aspergillus udagawae*, *Aspergillus viridinutans*, and *Aspergillus calidoustus*) are increasingly recognized and may account for up to 10–30% of IA cases [1,5]. These emerging species are clinically relevant due to intrinsic antifungal resistance and diagnostic challenges, often associated with refractory or relapsing IA [1]. From both epidemiological and clinical perspectives, aspergillosis plays a major role in immunocompromised populations. Patients with hematologic malignancies (especially acute leukemias) undergoing intensive chemotherapy or hematopoietic stem cell transplantation (HSCT), as well as solid organ transplant recipients, individuals with advanced HIV infection, or those receiving high-dose corticosteroids or novel immunosuppressive agents (e.g., targeted biologics), are at particularly high risk for IA [6]. In these groups, *Aspergillus* remains one of the leading causes of infectious mortality [1].

Importantly, severe viral respiratory infections such as influenza and COVID-19 can predispose even non-classically immunocompromised individuals to IA, particularly those with acute respiratory distress syndrome (ARDS). In such cases, IAPA and CAPA have been linked to poor outcomes, with mortality rates often exceeding 50% [2,3]. This evolving risk landscape has prompted updates in diagnostic definitions and prevention strategies. In 2020, the European Organization for Research and Treatment of Cancer and the Mycoses Study Group (EORTC/MSG) consortium revised the definitions of invasive fungal disease (IFD), expanding the list of host factors (e.g., solid organ transplantation, certain inborn errors of immunity such as STAT3 deficiency) and introducing new microbiological criteria (such as PCR testing on clinical specimens) to better identify probable aspergillosis cases [6,7]. These updated definitions [7] offer a broader framework to recognize IA beyond the classical population of hematologic patients, acknowledging that IA is now increasingly reported in critically ill patients without prolonged neutropenia (e.g., ICU patients with COVID-19) [8]. In parallel, guidelines have been developed for newly identified at-risk populations. For instance, specific diagnostic criteria for CAPA have been proposed during the pandemic [9], and antifungal prophylaxis recommendations have been formulated for patients with acute myeloid leukemia (AML) receiving novel targeted therapies, given the persistent high risk of IFD in this context [10].

Here, we will specifically address three major categories of immunocompromised hosts: people living with HIV, solid organ transplant recipients, and patients with hematologic malignancies or post-HSCT. In addition, we will examine viral-associated pulmonary aspergillosis, with a focus on influenza-associated pulmonary aspergillosis (IAPA) and COVID-19-associated pulmonary aspergillosis (CAPA), which have emerged as distinct clinical entities. Although these forms can also occur in patients with classical immunosuppression, they have been increasingly reported in critically ill patients without traditional risk factors, suggesting a broader at-risk population [11].

## 2. Microbiology, Transmission and Pathogenesis

Species in the filamentous fungal genus *Aspergillus* display a wide diversity of lifestyles and are of great importance to humans [12]. On the other hand, several *Aspergillus* species are known to cause human disease, including *A. fumigatus*, *A. flavus*, *A. niger*, *A. terreus*, *A. versicolor*, *A. ustus*, *A. lentulus*, and *A. nidulans* (also known as *Emericella nidulans*). These molds are ubiquitous in the environment and produce airborne conidia [13]. *Aspergillus* species differ not only in pathogenicity but also in morphological characteristics, including the size of their conidia, which may influence airborne dispersal and deposition in the respiratory tract.

The conidial size of *A. fumigatus*, typically ranges from 2.0 to 3.5 μm, favoring deep alveolar penetration [14,15]. In contrast, *A. flavus* conidia are generally 3.0–6.0 μm in diameter, slightly larger and more irregular in shape [16]; *A. niger* and *A. terreus* produce conidia of approximately 3.5–5.0 μm, which may deposit higher in the airways and cryptic species such as *A. lentulus* and *A. calidoustus* often resemble *A. fumigatus* macroscopically, but have slightly larger or more variable conidia (2.5–4.5 μm), requiring molecular identification for accurate differentiation [17]. These variations in size can affect the aerobiology and infectious potential of each species, with smaller conidia better suited to reach the terminal alveoli—a key factor in the pathogenesis of invasive pulmonary aspergillosis (IPA) [18]. Humans inhale hundreds to thousands of these infectious propagules on a daily basis. Due to their small size, *Aspergillus* spp. conidia can bypass mucociliary clearance and lodge in the lower respiratory tract. In immunocompetent individuals, the coordinated action of the respiratory epithelium, lung-resident macrophages, as well as recruited neutrophils and monocytes clear conidia efficiently [19]. The removal of fungi from an infected host depends on the rapid migration of a sufficient number of phagocytes to the site of infection and the recognition of fungal PAMPs via PRRs (both soluble and membrane-bound). This leads to the ingestion and, ultimately, the degradation of the ingested fungal cells within the phagocytes [20]. A key aspect of this process is the maturation of conidia, which, upon swelling, lose the thin hydrophobic RodA protein layer, a surface component of *Aspergillus* that masks the immunogenic components of the cell wall, such as β-glucan. This allows for increased exposure of fungal PAMPs, facilitating recognition by PRRs such as Dectin-1 and Dectin-2. Dectin-1, expressed on macrophages, neutrophils, and dendritic cells, recognizes β-glucan fragments on swollen conidia, mediating a prolonged inflammatory response and the activation of immune cells. Similarly, Dectin-2, expressed on macrophages and dendritic cells, binds fungal mannans, facilitating early recognition of conidia and the subsequent recruitment of phagocytes [21].

CR3 (CD11b/CD18), a pattern recognition receptor expressed on phagocytes, binds fungal chitosan and plays a critical role in orchestrating both phagocytosis and cytokine release in response to *Aspergillus* [22]. A key molecule in this early recognition phase is pentraxin 3 (PTX3), which enhances phagocytosis by binding to fungal surfaces [23]. Activated immune cells then release chemokines such as CXCL1 and CXCL5, which recruit neutrophils to the site of infection. Other inflammatory mediators—such as leukotriene B4 (LTB4), complement component C5a, and galectin-3—further amplify neutrophil recruitment and promote swarming behavior [24]. In addition to degranulation, phagocytosis, ROS generation, and cytokine production, pathogen-induced activation of neutrophils also initiates cellular processes to expel chromatin to the exterior for neutrophil extracellular trap (NET) release. NETs also appear to modulate host immunity to *A. fumigatus* through release of long pentraxin (PTX) 3 facilitating pathogen recognition [25,26].

Monocytes migrate into the alveolar space, where they differentiate into monocyte-derived dendritic cells and collaborate with neutrophils to engulf *Aspergillus* conidia. Within the phagolysosomes, fungal killing is driven by reactive oxygen species (ROS) generated by NADPH oxidase. The subsequent release of interleukins and other pro-inflammatory mediators further amplifies the immune response by recruiting plasmacytoid dendritic cells. These cells enhance the oxidative burst, promoting efficient conidial clearance from the lungs [27,28].

Cell-mediated immunity plays a crucial role in antifungal immunity or allergy-associated tissue damage. Fungal antigens are carried to lymph nodes, where they are processed and presented by antigen-presenting cells, leading to the differentiation of naive CD4^+^ T cells into distinct T-helper subsets. In the context of invasive aspergillosis, the functional polarization of these T cells is critical in determining the outcome of infection. Th1 cells, characterized by T-bet expression and the production of interferon-gamma, promote macrophage activation [29]. A robust Th1 response is generally associated with protective immunity, and its deficiency correlates with increased susceptibility to IA.

For the Th17 cells, the differentiation is driven by innate cytokines such as IL-1α, IL-1β, IL-6, and IL-23, which activate the lineage-defining transcription factor RORγT (retinoic acid receptor-related orphan receptor gamma T). These Th17 cells secrete IL-17A and IL-22, promoting neutrophil recruitment and strengthening epithelial barrier defenses through the induction of antimicrobial peptides. While Th17 responses contribute to fungal control in the early phases of infection, excessive or prolonged activation may lead to immunopathology and tissue damage. Th2 responses, characterized by IL-4, IL-5, and IL-13 production, are considered detrimental in IA, as they interfere with protective Th1 activity and may impair antifungal effector mechanisms.

Regulatory T cells (Tregs), defined by Foxp3 expression and the secretion of IL-10 and TGF-β, act to restrain excessive immune activation and limit tissue damage. However, in the setting of IA, an overabundance or overactivity of Tregs may suppress effective Th1 and Th17 responses, contributing to fungal persistence. These observations highlight the necessity of a balanced adaptive immune response in IA—one that ensures pathogen clearance while preventing immune-mediated injury [30,31]. Fungal antigens are carried to lymph nodes, where T helper cells differentiate into Tbet+ T helper 1 cells. These cells then travel to the lungs to produce IFN-γ, activating macrophages antifungal functions. Meanwhile, T helper 2 cells, which secrete IL-4, IL-5, and IL-13, are implicated in allergic responses. Additionally, T helper 17 cells, through their secretion of IL-17, contribute to certain antifungal activities [29].

## 3. Immunocompromised Patients

### 3.1. Hematologic Patients

The survival rates of patients with hematological malignancies have seen significant improvements due to advancements in medication and supportive therapies, which have also led to a rise in ICU admissions [32].

These patients are more susceptible to developing serious complications, one of which is invasive fungal infections (IFI) [33]. IFIs are common in bone marrow transplantation (BMT), with an incidence ranging from 5% to 20%. Risk factors include the development of graft-versus-host disease, increased human leukocyte antigen mismatch, and co-infection with respiratory viruses or cytomegalovirus (CMV). IA represents the most prevalent IFI among recipients, accounting for 43% of cases, with *A. fumigatus* being the predominant species [34].

The diagnosis of IFI in hematology patients is associated with the highest mortality [35].

In Europe, 3.7% of ICU patients had proven or probable IA with an ICU mortality of 87–97% in immunocompromised patients [36,37] while the incidence among patients with leukemia is almost 12.7%, but the use of antifungal prophylaxis in high-risk patients has modified this data [38,39].

All blood malignancies involve the immune system, which promotes the IFI development for T cell dysfunction, hypogammaglobinemia, neutropenia due to bone marrow involvement, and leukopenia due to chemotherapy or BMT.

As seen before, neutrophils are the first defense line against fungi: an absolute neutrophil count < 500/mm^3^ predisposes to infections as much as an average count but immature neutrophils (acute myeloid leukemia) [40,41]. The most critical risk factor for IA is prolonged and profound neutropenia. Neutropenia is frequent in patients with acute leukemia or myelodysplastic syndrome (MDS) during remission induction chemotherapy and in patients undergoing allogeneic HSCT. In the former, the period more susceptible to IFI is the time until the first remission. In contrast, the next high-risk period in the latter group follows the conditioning chemotherapy regimen.

A deficiency in lymphocyte production and trafficking and alterations in lymph organ function, second line of defense against fungi, promote IFI in lymphocytic leukemia involving innate and adaptive immunity [42,43].

In HSCT patients also, the type of immunosuppression can promote an impairment of the host’s defense, and after neutropenia resolution, the development of a graft-versus-host disease (GvHD) may require escalation of immunosuppression therapy, increasing the risk of IFI. Other significant risk factors include renal or liver dysfunction as well as age over 50 years, and they represent the strongest mortality predictors among patients with acute myelogenous leukemia (AML) post induction chemotherapy [44].

Signs or symptoms are not so specific in an early phase. The challenge in the management of IA in hematologic patients is to early identify high-risk patients through host risk factors, clinical criteria and microbiological criteria to make a presumed diagnosis to start therapy and reduce mortality. A defined diagnosis can be made through autopsy or with a positive culture of *Aspergillus* in a specimen obtained by biopsy or needle aspiration from a sterile site that is clinically or radiologically abnormal in which hyphae or melanized yeast-like forms are present.

The gold standard for diagnosing IA remains culture, as it allows precise identification of the *Aspergillus* species, thanks to advancements in identification techniques, such as the introduction of MALDI-TOF assay, as well as antimicrobial susceptibility testing of the isolate. However, the diagnosis of IA is still challenging for several reasons: tissue sampling may be difficult or contraindicated in patients with hemodynamic instability, thrombocytopenia, or coagulation disorders; the sensitivity of cultures is often low and may vary depending on the type and quality of the specimen, the timing of collection, and the prior use of antifungal agents [45,46].

For this reason, indirect methods must be introduced, such as detecting galactomannan (GM) or β-D-glucan (BDG) in serum, bronchoalveolar lavage (BAL), or polymerase chain reaction (PCR) assay [7].

A recent systematic review by Bukkems et al. [47] describes GM antigen assay role in invasive pulmonary aspergillosis (IPA) diagnosis in hematological patients, including all IPA categories. Considering the type of patients, the type of sample analyzed, and the IPA categories is essential to establish the sensitivity and specificity of GM assays. The results show that for serum and BAL, an optical density index (ODI) of 0.5 and 1.0, respectively, provides the best diagnostic accuracy with a sensitivity and specificity different for each IPA category (Table 1).

Without treatment, mortality for IA is 100% in these patients, while with antifungal treatment the overall mortality is between 30% and 40% for AML and 60% for HSCT patients [48].

### 3.2. HIV Patients

In the absence of treatment, HIV infection can cause marked deterioration of immune function. The degree of immune suppression is directly linked to the emergence of opportunistic infections, particularly when circulating CD4+ lymphocytes are below 200 cells/mm^3^. Among these infections, fungal diseases such as pneumocystosis, cryptococcosis, histoplasmosis, and aspergillosis contribute significantly to morbidity and mortality in individuals with AIDS [49].

Fungi belonging to the *Aspergillus* genus are responsible for a range of clinical syndromes with varying degrees of severity in humans [50]. Individuals with advanced stages of HIV are vulnerable to different manifestations of aspergillosis, including invasive forms, atypical bronchial obstructions, and chronic pulmonary involvement. Although the lungs represent the primary site of infection, dissemination to other organs such as the heart, paranasal sinuses, kidneys, and central nervous system can also occur [51]. Superficial and allergic forms, such as nail infections, keratitis, and outer ear involvement, are relatively uncommon in this population, although some studies suggest keratitis may be slightly more frequent in HIV-infected individuals [52].

In contrast to patients with neutropenia or those undergoing transplantation, in whom the disease typically presents acutely, people with HIV (PWH) often experience more protracted, subacute forms of infection. Nevertheless, both the acute and chronic presentations may carry a significant risk of death [53]. The frequency of aspergillosis in PWH has declined since the introduction of effective antiretroviral therapy, but the infection still occurs [54,55]. An analysis of over 35,000 patients from a U.S. national database reported an incidence of 3.5 cases per 1000 person-years [56], consistent with other large-scale studies reporting incidence rates around 0.43% [54,55]. Neutropenia and corticosteroid therapy were present in nearly half of all reported cases [53,55], and other contributing factors include concurrent *Pneumocystis jirovecii* infection and CD4+ counts below 50–100 cells/mm^3^ [57,58,59,60,61,62,63].

In their review, Doumbo et al. [64] described the epidemiological context in Mali, where health systems are under strain due to high tuberculosis prevalence and a moderate HIV burden. The estimated population of Mali was 21,251,000 in 2020 (UN), of which 45% were children < 14 years of age. Among HIV-positive individuals, the less frequently reported invasive fungal infections included IA (*n* = 1230), keratitis (*n* = 2820), bloodstream *Candida* infections (>1060), and mucormycosis (*n* = 43). These numbers are thought to underestimate the true burden of fungal infections due to insufficient access to diagnostic resources.

A study by Truda et al. [65], involving 25 HIV-positive patients (aged 23–58 years), identified 11 cases of invasive and 14 cases of chronic pulmonary aspergillosis. The overall prevalence in their cohort of nearly 20,000 individuals was 0.1%. Most patients with the invasive form had CD4+ counts below 100 cells/mm^3^, whereas over 40% of those with chronic disease had counts above 200 cells/mm^3^. A history of tuberculosis was common, particularly among patients with chronic aspergillosis (85.7%). The diagnosis was generally supported by radiological findings and culture positivity (71.4%). Crude mortality was high: 72.7% in the invasive group and 42.8% in the chronic group. Despite its rarity (0.1% in PWH), IA should be suspected in immunocompromised patients with unexplained pneumonia refractory to conventional therapy, and chronic pulmonary disease should be considered in those with respiratory decline and a previous history of tuberculosis.

In a systematic review, Yerbanga IW et al. [66] confirmed the presence of IA in Africa, highlighting a substantial number of undetected cases. The review emphasized key differences in epidemiology between African countries and high-income settings and called for more targeted surveillance in regions with growing numbers of at-risk individuals.

Denning et al. [67] analyzed 54 studies and identified 859 HIV/AIDS patients with aspergillosis. Among the 853 with available outcome data, 707 had died, corresponding to a mortality rate of 83%. The majority of reported cases over the past 15 years occurred in untreated individuals, those with poor adherence to antiretroviral therapy, or in advanced stages of immunosuppression, particularly when CD4+ counts were under 50 cells/mm^3^. Although aspergillosis is no longer listed among AIDS-defining conditions, its high fatality rate in this population underscores the importance of considering it in patients presenting with fever of unknown origin and/or lung infiltrates, particularly when cavitary lesions are present.

### 3.3. Organ Recipients

Solid organ transplantation (SOT) has significantly improved the lives of individuals with end-stage organ failure. Despite the success, SOT recipients are at a higher risk of opportunistic infections due to long-term immunosuppression, with IA being a notable cause of morbidity and mortality. The management of IA in SOT recipients remains challenging despite advancements in transplant medicine and antifungal therapies.

A study conducted on a cohort of 960 cases with probable or proven IA, as reported in the Prospective Antifungal Therapy Alliance (PATH Alliance) registry, revealed that 48.3% of the individuals had hematologic malignancies, 29.2% were recipients of SOT, 27.9% had undergone HSCT, and 33.8% were neutropenic. Among these patients, the prevalent clinical manifestations of IA included IPA and rhinocerebral aspergillosis [68].

Despite a decline in mortality rates among transplant recipients in recent years from 65–92% to 22%, it is estimated that approximately 9.3–16.9% of all deaths occurring within the first-year post-transplantation can still be attributed to IA [69].

### 3.4. Liver Transplant Recipients

Liver transplant (LT) recipients are at risk for IA, with a prevalence of 1.8%, according to recent studies [68,70]. This population’s mortality rate associated with IA ranges from 50% to 100%. However, there has been a significant decrease in both the incidence and mortality rates over the past decades, with the incidence dropping from 10% to 1.8% and mortality declining from 100% to approximately 50% [68,70,71].

Despite these improvements, the mortality rates among LT recipients remain higher compared to other transplant procedures [72]. IA manifests earlier in LT recipients than in other SOT recipients, excluding the heart. The diagnosis is typically made within 17 to 128 days post-transplantation, whereas for all SOT recipients the median time to diagnosis is around 100 days [69,73,74,75].

Risk factors for IA in LT recipients include immune system dysregulation induced by bacterial translocation, complement system insufficiency, monocyte suppression, and impaired neutrophil phagocytosis. Management of LT recipients involves reduced levels of immunosuppression, with some individuals potentially achieving immunological tolerance. Corticosteroids have been identified as a risk factor for IA, while calcineurin inhibitors exhibit activities against *Aspergillus* species [76,77,78].

Other risk factors for IA in LT recipients include pre-transplant hepatic failure, primary allograft failure or dysfunction, re-transplantation, post-transplant dialysis, and elevated transfusion needs. CMV infection has been associated with a higher likelihood of IA occurring beyond 100 days post-transplantation, and a MELD score exceeding 30 is linked to a heightened risk of developing IFIs [73,79,80,81]. Steroid boluses administered before surgery were linked to an increased occurrence of IPA after transplant operations, with a rate of 16% versus a mere 0.4% [70]. Additionally, a higher incidence of IFIs was observed in patients who received corticosteroids or antibiotics within three months before surgery, showing a significant difference (64% vs. 34% for corticosteroids and 61% vs. 31% for antibiotics) [82]. Factors such as surgical durations longer than 11 h, extensive red blood cell transfusions during surgery (with an odds ratio of 23.1), and plasma and platelet transfusions were strongly associated with an increased risk of IFIs. Surgical procedures such as [83,84,85] choledochojejunostomy and Roux-en-Y anastomosis were also associated with a significantly higher risk of IFIs, two-fold and 16-fold, respectively [69,86]. In a comprehensive study of 1730 patients undergoing LT, with subsequent development of CNS lesions, 11 were found to have lesions due to infections. *A. fumigatus* was the most common agent, typically appearing about 21 days after transplantation, with cases ranging from 10 to 27 days post-operation [87].

### 3.5. Renal Transplant Recipients

The occurrence of IA in individuals who have received a kidney transplant (KT) is reported to be between 0.7% and 4% [88,89]. Factors that elevate the likelihood of IA within this group encompass the use of high doses of methylprednisolone, prolonged therapy with corticosteroids, and the necessity for hemodialysis due to transplant failure [90].

Following SOT, particularly in the initial six months, there is a heightened use of drugs that suppress the immune system. This period is notably vulnerable to opportunistic infections, including IPA [91]. Despite this, recipients of KTs typically undergo a continuous treatment regimen that includes corticosteroids, calcineurin inhibitors, and anti-proliferative agents [92]. Although KT recipients have a comparatively lower incidence of post-transplant IPA than recipients of other transplants, the extension of KT survival has increased in older patients undergoing prolonged intensive immunosuppressive treatment [93].

Although the risk of post-transplant IPA is lower in KT recipients compared to other transplant recipients, advances in the longevity of KT recipients have led to an increasing number of elderly patients receiving prolonged high levels of immunosuppression [94]. It has also been documented that significant risk factors for the development of IA after KT include the receipt of methylprednisolone in doses greater than 3 g, prolonged use of corticosteroids, and the need for hemodialysis following transplant failure [95,96].

### 3.6. Lung Transplant Recipients

In recipients of lung transplant (LT), IA emerges as the most common fungal infection. Historically, the occurrence of this infection in LT patients was estimated to be between 4% and 23%. However, recent research indicates a decline in its prevalence [97]. The time from LT to the development of IA has increased, a change attributed to the use of antifungal prophylaxis early after transplantation. This has led to a shift in the median time of onset of the infection from 120 days to 483 and 508 days post-transplant, respectively [98].

*A. fumigatus* is the primary fungal species responsible for IA in these patients. A study tracking 900 adult LT recipients in various international centers for four years found that 79 individuals had 115 cases of infection. Another study of 251 patients showed that *Aspergillus* was present in 86 (33%) cases, ranging from colonization to tracheobronchitis and IA [98].

The significance of *Aspergillus* spp. colonization before LT remains controversial. Some researchers emphasize its importance, particularly during the first-year post-transplant and when colonization occurs within six months prior to surgery. However, other studies do not support a clear link between pre-transplant colonization and post-transplant IA [99,100].

Other identified risk factors for IA in LT recipients include bronchial anastomotic leaks, surgical site complications, airway narrowing, allograft dysfunction or ischemia, reperfusion injury, CMV infection, bronchiolitis, and the need for increased immunosuppression to prevent graft rejection [100].

Ischemic damage at the bronchial anastomotic site can lead to ulcerative tracheobronchitis, a form of IA, and potentially bronchovascular fistulas, which are associated with serious bleeding risks [69,100]. Mortality rates for LT recipients with IA vary significantly, from 23% to 29% in tracheobronchitis cases to 67–82% in severe pulmonary infections. Nonetheless, some data suggest the mortality rate could be as low as 20%. Patients with central nervous system involvement or disseminated disease generally have a poorer prognosis [87].

### 3.7. Heart Transplant Recipients

The incidence of IA among heart transplant (HT) recipients is reported to be between 1% and 14%, with the risks for developing this condition differing based on whether it occurs early or later after the transplant. Factors such as undergoing hemodialysis, needing thoracic surgery again, and increased levels of immunosuppressive drugs are linked to a higher likelihood of developing IA among these patients. Not all patients with IA will show signs of neutropenia, making it essential to diagnose the condition through clinical observation and imaging results. The most common presentation of IA in this group is IPA, particularly within the first three months following a HT. On the other hand, cases of IA that arise later tend to show a wider spread of the disease, including to the CNS and other areas outside the lungs [100,101,102]. Research by Montoya and colleagues revealed that HT patients with either form of aspergillosis did not have neutropenia. This underscores the importance of considering IPA in patients who, despite not having neutropenia, exhibit fever and respiratory symptoms, have positive cultures from respiratory secretions, and show unusual patterns on lung scans, such as nodules, especially within three months after their transplant [103].

A 2006 study highlighted a one-year post-diagnosis mortality rate of 67% among HT patients with IA [104]. However, more recent data suggest a reduction in mortality to 38%, indicating progress in managing this condition in the transplant population [102].

## 4. Coinfections

### 4.1. COVID-Associated Pulmonary Aspergillosis (CAPA)

Historically, the development of pulmonary aspergillosis was mainly linked to specific conditions of profound immunosuppression, such as hematologic malignancies, stem cell transplantation, cytotoxic therapies, or prolonged use of corticosteroids and other immunosuppressants. More recently, the spectrum of risk factors has widened to include inherited immunodeficiencies and reduced lymphocyte subsets [58,105].

The COVID-19 pandemic has introduced new clinical scenarios. Infection with SARS-CoV-2 is now recognized to induce complex immune dysregulation, affecting both T helper 1 and 2 responses [106,107,108]. Although a direct inhibitory effect on antifungal host defense has not been definitively proven [109], there is growing concern that either the infection itself or the immunomodulatory treatments commonly used—such as corticosteroids and biological agents—may facilitate opportunistic fungal infections. This has prompted the identification of a specific form of aspergillosis emerging in COVID-19 patients, termed COVID-19-associated pulmonary aspergillosis (CAPA) [9]. However, evidence on the prevalence and relative contribution of different risk factors in this setting remains limited. Notably, in patients with COVID-19-associated acute respiratory distress syndrome (ARDS) and concomitant severe infections, mortality has been reported to increase by 16–25% when aspergillosis is present [110,111].

Montrucchio et al. highlighted the relevance of prolonged respiratory support, both invasive and non-invasive, which is commonly required in severe COVID-19, as a potential risk factor for CAPA [112]. Additional attention should be given to the possible role of corticosteroids and immunobiological therapies in increasing both incidence and mortality. Furthermore, the endothelial damage caused by SARS-CoV-2 [113] and its effects on the host immune response [114] may contribute to increased susceptibility to fungal infections.

Environmental and logistical challenges in ICUs during the COVID-19 pandemic—such as high patient volume, altered ventilation systems, structural renovations, and extended use of isolation rooms—might also have played a role in promoting fungal colonization and infection.

Chavda et al. reported a high prevalence of opportunistic fungal infections among COVID-19 patients, particularly in the elderly population [115]. Data from the UK indicate that individuals aged between 55 and 81 years represent the highest proportion of coinfected patients [116], while a Spanish study found the median age to be 62 years (range 48–74) [117]. In a retrospective cohort from a tertiary care center in North India, Paul et al. documented that COVID-19 patients with diabetes mellitus were at significantly increased risk of developing angio-invasive fungal infections [118]. A prospective study conducted at the India Institute of Medical Sciences (AIIMS), in New Delhi, on ten patients with subacute IPA, all with diabetes and steroid exposure, described cough—often preceded by hemoptysis—as a typical symptom [119].

The diagnosis of CAPA remains challenging, primarily due to its clinical and radiological resemblance to severe COVID-19 pneumonia. Moreover, conventional diagnostic markers such as serum galactomannan may show reduced sensitivity in non-neutropenic patients, as neutrophils can clear fungal antigens from the bloodstream before detection [120]. This has led to a disease course that, unlike classical angio-invasive aspergillosis in neutropenic patients, is often initially limited to airway invasion for several days before vascular dissemination occurs.

Two years after the onset of the pandemic, numerous epidemiological and observational studies continue to emphasize the clinical impact of SARS-CoV-2 in predisposing patients to secondary fungal infections. The emergence of these coinfections appears to be strongly influenced by the use of mechanical ventilation, intravascular catheters, and immunosuppressive therapies [121].

### 4.2. Influenza-Associated Pulmonary Aspergillosis (IAPA)

Influenza-associated pulmonary aspergillosis (IAPA) is an emerging complication of influenza infection, often associated with *Aspergillus* tracheobronchitis [122,123]. It has emerged as a significant complication in critically ill patients, particularly those admitted to ICUs with severe influenza, markedly increasing influenza-associated mortality. Traditionally considered a concern limited to severely immunocompromised hosts, recent evidence shows that IAPA can occur in non-neutropenic patients without classical risk factors, expanding the at-risk population [11,124]. Recent cohort studies and meta-analyses estimate the incidence of IAPA in ICU influenza patients to range between 10% and 20%, with mortality rates frequently exceeding 50% [2,11,122,125]. In a large retrospective multicenter cohort by Schauwvlieghe et al., 19% of influenza ICU patients developed IAPA, with a 51% mortality rate compared to 28% in those without aspergillosis [11]. Another recent Swiss study confirmed a prevalence of ~11%, with early onset (within 72 h of ICU admission), highlighting the aggressive and rapidly progressive nature of IAPA [124,125]. Pathophysiologically, influenza virus causes extensive damage to the respiratory epithelium, disrupts mucociliary clearance, and impairs local immune responses, creating a permissive environment for *Aspergillus* invasion. In addition, the viral infection induces dysregulation of innate immunity, particularly of neutrophil and macrophage function, while promoting excessive interferon-γ responses, which paradoxically impair antifungal activity, resulting in a defective Th17-immune response, depletion of macrophages, and impaired killing of *Aspergillus* conidia by macrophages [126]. However, recent experimental evidence suggests that restoration of IL-17 signaling alone is insufficient to ensure adequate fungal clearance during IAPA, indicating the involvement of additional immunological impairments contributing to defective antifungal immunity [126]. Supporting this hypothesis, Garcia et al. demonstrated in a murine model of IAPA that inhaled immunotherapy with pattern recognition receptor (PRR) agonists effectively reactivated innate immune responses, reversed profound immune paralysis, and improved *Aspergillus* clearance and survival [127,128]. A growing body of evidence identifies asthma and inhaled corticosteroid use as independent risk factors for IAPA, even in the absence of systemic immunosuppression [125]. Chronic pulmonary disease (e.g., COPD), male sex, and systemic corticosteroids further increase susceptibility [122,125]. Diagnosis remains challenging, as clinical and radiological findings of IAPA often overlap with those of viral pneumonia. Bronchoscopy with bronchoalveolar lavage (BAL) is recommended when feasible. Galactomannan (GM) detection in BAL fluid shows higher sensitivity than serum GM in IAPA patients [122]. Fungal culture, PCR, and β-D-glucan may provide supportive evidence. Management includes early antifungal therapy, most commonly with voriconazole or isavuconazole. Empirical antifungal treatment is often warranted in high-risk patients with persistent or worsening respiratory failure. However, data on prophylaxis remain inconclusive: the POSA-FLU trial assessing posaconazole prophylaxis in influenza ICU patients showed limited benefit, suggesting the need for better risk stratification [129]. Outcomes remain poor, especially when diagnosis is delayed or antifungal therapy is not promptly initiated. Mortality rates are significantly higher in IAPA patients compared to influenza patients without fungal co-infection, underscoring the need for heightened clinical awareness, early diagnostic workup, and multidisciplinary management.

## 5. Diagnosis

Because of the paucity of clinical signs and the limited sensitivity and/or specificity of radiology and mycological tests, the diagnosis of IPA is graded according to a scale of probability (possible, probable or proven) of disease [46,130]. The first version of definitions of invasive fungal infections (IFIs) for clinical and epidemiological research published in 2002 by a consensus group of the EORTC/MSG, refers to immunocompromised patients with cancer and HSCT [131]. In 2008, they had an update [132] in which data were insufficient to establish a threshold for *Aspergillus* GM, and the diagnostic role of 1,3-beta-D-glucan (BDG) was uncertain.

It is pertinent to mention that the updated 2019 European Organization for Research and Treatment of Cancer/Mycoses Study Group Education and Research Consortium (EORTC/MSGERC) consensus definitions for invasive fungal disease (IFD) now include: (i) *Aspergillus* PCR and (ii) the molecular detection of fungal (*Aspergillus*) DNA in tissue, as diagnostic criteria for IA [45].

The 2020 Committee’s update tried to incorporate all these new findings [7,133,134] (Figure 2).

For IFI, as well as for aspergillosis, the definitions assigned three levels of probability to the diagnosis: for proven infection, it is necessary to confirm the presence of fungi in a sterile material during the microscopic exam with alterations in tissue structure as illustrated in Figure 3.

For probable infection, the presence of at least one host factor, one clinical criterion and one mycological criterion is required, while the absence of mycological criteria classified the case as possible infection.

It is important to underline that these guidelines refer to the immunocompromised population and not to the subset of critically ill patients in the ICU who do not fulfil the EORTC/MSG host factors [45].

The population of non-neutropenic, critically ill adult patients is highly heterogenous, including medical and surgical patients, with a wide range of baseline comorbidities and predisposing conditions for IFD (i.e., chronic obstructive pulmonary disease, long-term steroid therapy, hepatic cirrhosis, dialysis, near drowning, or diabetes, sepsis due to bacterial, viral, or parasitic agents, and severe postsepsis immunoparalysis) [68,133]. To address this diagnostic gap, alternative algorithms have been developed. In 2012, the AspICU, and subsequently in 2021, the Modified-AspICU (M-AspICU) algorithm, were proposed to improve IA diagnosis in the ICU settings by substituting the host factors with *Aspergillus*-positive lower respiratory tract specimen culture [7,133,134]. The AspICU criteria consider a combination of clinical signs (such as fever or respiratory deterioration), abnormal imaging, microbiological evidence (including positive culture for *Aspergillus* spp. from lower respiratory tract samples), and host factors not limited to classical immunosuppression [7,133,134].

More recently, to overcome the limitations of AspICU and to further refine the diagnostic classification in ICU settings, the Invasive Fungal Diseases in Adult Patients in ICU (FUNDICU) project was conceived with the aim of developing a standard set of definitions for IFD in non-neutropenic, ICU patients outside the classical immunocompromised patient populations, which could improve the generalizability and comparability of research results [68]. The FUNDICU proposes a more nuanced categorization of fungal infections in ICU patients into three levels: proven, probable, and possible IFD, based on a combination of clinical, radiological, and microbiological findings. Unlike the AspICU algorithm, FUNDICU incorporates advanced diagnostic tools such as galactomannan, β-D-glucan, and PCR from respiratory or blood samples, and recognizes viral coinfections (notably influenza and SARS-CoV-2) as host risk factors for probable IFD [68]. By adapting the diagnostic criteria to the reality of ICU patients, FUNDICU aims to facilitate early recognition of fungal infections and improve the standardization of clinical research endpoints in this high-risk population (Table 2).

The gold standard for diagnosing aspergillosis is a sterile sample biopsy obtained through bronchoscopy, surgical procedure, or trans-thoracic needle. These procedures can have some executive limitations due to hemodynamic instability, coagulation disorders, or respiratory failure of these patients [124,133,134].

For this reason, the diagnosis should start with the clinical evaluation of patients (host factors, symptoms, and signs of presentation), followed by the execution of diagnostic tests to confirm or not the probability of having aspergillosis.

Usually, clinical presentation involves the pulmonary system with cough, respiratory distress, and dyspnea. Fever and resistance to antibiotics is another sign that is not specific. Chest pain or hemoptysis are frequent in case of angio-invasive growth, which can promote the dissemination to other organs: CNS with convulsions, meningitis, and other clinical manifestations with a bad prognosis [135]. A new radiologic finding or persistent chest infiltrates despite antibiotics should raise suspicion of a superimposed fungal infection.

The role of imaging is to detect typical features to make an early diagnosis of IPA, evaluate the progression, and detect complications. Computed tomography (CT) is the preferred method of diagnosis and follow-up. A non-contrast CT was performed initially, but the contrast is necessary to detect the angio-invasion in the lesion. Radiologic presentation varies about the immune status of patients. Specifically, pulmonary nodules can be observed in both neutropenic and non-neutropenic patients, with a distribution reflecting the infection’s behavior (airway invasive or angioinvasive). These nodules may be associated with a halo sign, which, though not specific, serves as an early radiological indicator, particularly in the presence of host risk factors such as neutropenia. The halo sign has a high positive predictive value for diagnosing IPA and is associated with a favorable prognosis. The cavitation or air crescent sign appears later in the infection and is indicative of immune recovery. Other radiologic features linked to pulmonary nodules include hypodense signs and transfissural extension. Consolidation, with or without the reverse halo (atoll) sign or vascular occlusion sign, is another pattern observed in neutropenic patients, though it is not exclusively associated with aspergillosis [136,137,138].

GM is a polysaccharide in the cell wall of *Aspergillus* spp., incorporated into the wall during fungal growth and released into the circulation when fungus invades the endothelial compartment. There are some differences between the release of GM in neutropenic hosts versus glucocorticoid-treated hosts. In the former, the infection shows an extensive angio-invasion and a high fungal burden; in the latter, the opposite has a significant inflammation role. This evidence explains the differences in the circulating GM amount in these two groups: high in neutropenic and low in steroid-suppressed patients. Platelia *Aspergillus* sandwich enzyme immunoassay is used to detect GM: a monoclonal antibody derived from rats binds to the GM molecule, so the presence of the antigen in the specimen produces a monoclonal antibody-GM-monoclonal antibody complex and the positivity of the test is revealed by the addition of a chromogenic substrate. The value is expressed as ODI.

A positive serum result should be based on a cut-off GM index ≥ 0.5 after testing two separate serum samples or a single sample with a value of ≥1.0 [139].

A positive result on BAL fluid should be based on a cut-off GM index ≥ 1.0 after testing two aliquots of a single BAL fluid sample [140]. GM testing sensitivity and specificity depend on the cut-off level used, patient population, and immune status of the host; in particular, its sensitivity in neutropenic patients is higher than in ICU patients without traditional risk factors. Anti-mold prophylaxis/treatment can reduce the sensitivity of the GM assay. It has a prognostic role with a reduction in its value from 7 to 14 days. False negatives are possible in patients treated with antifungal therapy. At the same time, false positives are feasible in patients with another fungal infection. It can be examined on plasma or BAL, and it demonstrates a relatively high negative predictive value (>90%) with a relatively low positive predictive value (<50%) [140].

Results have shown that patients without clinical signs have tested positive for GM, including those who were taking piperacillin/tazobactam, beta-lactams, and other antibiotics. Additionally, patients who were receiving total parenteral nutrition or crystalloid solutions also had false-positive GM results. Furthermore, patients with mucositis or an altered intestinal barrier displayed false-positive GM results due to dietary factors or glucose-containing solutions [141].

1,3-BDG is a cell wall component; it is not specific for *Aspergillus* but is released during IFI. There are many assays, but the guidelines suggest the use of Fungitell^®^. This assay uses a modified pathway in the *Limulus* Amebocyte lysate (LAL) test by removing factor C from LAL; in the absence of factor C, the coagulation cascade is activated only by the presence of 1,3-BDG; the activity can be measured with a colorimetric method. The positivity is established by a cut-off of ≥80 ng/L detected in at least two consecutive serum samples. Another assay, such as Dynamiker^®^ Fungus (1-3)-β-D-glucan assay (DFA), uses a cut-off lower than Fungitell, 20 pg/mL. Many studies investigated how accurately BDG tests can detect IFI in ICU patients. The results show that there is a lot of variation in how sensitive and specific the tests are. In ICU patients, false positive results can occur due to various clinical factors and conditions such as the use of surgical gauzes, renal replacement therapy, albumin transfusion, and broad-spectrum antibiotics. The specificity and PPV of the test can be improved by conducting 2 consecutive BDG tests, without significantly impacting the NPV. Additionally, combining BDG testing with other fungal biomarkers, such as *Candida albicans* germ tube antibody (CAGTA), or using clinical prediction rules, like the *Candida* score, may enhance the test’s effectiveness. As the NPV is generally high, a negative BDG result can be used to stop or not start empirical antifungal therapy. However, the role of a positive BDG result in prompting pre-emptive antifungal strategies is less clear because of the relatively low PPV [142].

False positives are frequent in patients receiving albumin, hemodialysis with cellulose membranes, intravenous immunoglobulin and intravenous amoxicillin-clavulanic acid.

PCR-based methods could be applied to any specimen, showing high sensitivity and specificity for detecting *Aspergillus* DNA. There are several assays with a comparable level of sensibility and specificity. Sensitivity is lower in serum samples than in respiratory specimens, and performance can be reduced in patients with antifungal prevention or treatment. They are also different in the type of population examined. The MycAssay *Aspergillus*^®^ and the AsperGenius^®^ assay are recommended for PCR detection of *Aspergillus* spp. DNA in respiratory samples. These assays are helpful in the case of antifungal resistance in *Aspergillus* spp. They show a sensitivity of almost 78% and a specificity 100% [143].

A novel lateral flow device using an *Aspergillus*-specific monoclonal antibody represents a point-of-care diagnosis of IA. It is quick and cheap. It detects an extracellular glycoprotein secreted by *Aspergillus* spp., during active growth. Serum and BAL are the samples, but it is possible to have a cross-reactivity with *Penicillium*. The main diagnostic tests for aspergillosis, excluding culture-based methods, are summarized in Table 3.

Novel biomarkers or targets based on IA pathogenesis have been studied to discriminate *Aspergillus* infection from those without disease. The rationale is the expression of immune receptors by immune cells involved in the first line of defense against inhaled fungal spores. Cytokine profiles in BAL samples from IPA patients were compared to the control, and the best cytokine was IL-8 with elevated sensitivity (90%), specificity (73%) and NPV (88%) [144].

Another biomarker is triacetyl-fusarinine C (TAFC), a fungal molecule produced by some molds. One of these is *A. fumigatus*. It is a secreted siderophore that can be used as a biomarker. In fact, its detection in bodily fluids (e.g., serum, BAL or urine) may aid the diagnosis of IA and represents a promising tool to support the early identification of IPA in high-risk patients; if it has been combined with GM-BAL, the sensitivity and specificity of GM increased [145], but other studies have shown it has no benefit for IPA diagnosis, so its role is controversial. However, TAFC can be measured in urine samples with mass spectrometry.

In case of suspected fungal pneumonia, *Aspergillus* volatile organic compounds, secondary metabolites of varied chemical nature, can be detected in breath: α-trans-bergamotene, β-trans-bergamotene, β-vatirenene-like sesquiterpene or trans-geranyl acetone identified in IPA patients with a sensitivity and specificity of 94% and 93%, respectively [146]. Other studies are necessary to define their roles in diagnostics.

Proteome analysis of BAL shows host and fungal proteins expressed during IPA, which in the future can represent novel biomarkers in *Aspergillus* diagnosis [147].

## 6. Antifungal Prophylaxis and Pre-Emptive Treatment Versus Antifungal Therapy

The rising number of immunocompromised individuals has led to an increased risk of IFIs, which are associated with a high mortality rate due to various complications. The development of novel chemotherapy agents and immunosuppressive drugs has improved outcomes but also increased susceptibility to IFIs [148].

All of these factors must be taken into account when assessing a high-risk patient, in order to establish a treatment plan that selects the appropriate therapeutic regimen. This includes choosing the right drug based on its potential drug-drug interactions, efficacy, tolerability, and toxicity, while taking into account emerging pharmacological resistance and the introduction of drugs targeting novel molecular and immune pathways [10,149].

In certain cases, antifungal prophylaxis may be considered for patients without signs or symptoms of fungal disease. It is essential to balance the risk of drug-induced side effects and resistance with the potential benefits in terms of increased survival rates. Pre-emptive treatment involves administering antifungal therapy to a population based on laboratory tests that indicate early signs of IFD. Empiric treatment, on the other hand, is administered to patients at risk of fungal infection who present with fever as a sign or symptom of IFD [10].

Different antifungal treatments are available against *Aspergillus* spp. Polyenes such as amphotericin B deoxycholate (AmB) were the first agent substituted by triazoles for their superiority and less toxicity [150]. Initially, voriconazole proved to be superior to amphotericin B in the treatment of IA, also showing fewer side effects [150]. Subsequently, among azoles, isavuconazole demonstrated non-inferiority compared to voriconazole [151], with fewer adverse events, a better pharmacokinetic profile. Finally, posaconazole also showed non-inferiority to voriconazole [152]. Moreover, lipid formulations of AmB, which are less nephrotoxic, are an alternative when voriconazole and isavuconazole are not tolerated or contraindicated. Voriconazole and isavuconazole are available in oral (po) and intravenous (IV) formulations. They represent the primary regiments in IA therapy. A loading dose is necessary for voriconazole: 6 mg/kg IV every 12 h for two doses, followed by 4 mg/kg every 12 h. Therapeutic drug monitoring (TDM) is recommended due to its complex metabolism and numerous drug—drug interactions (DDIs), primarily due to its inhibition of CYP450 enzymes. These interactions can significantly alter the plasma concentrations of both voriconazole and co-administered drugs, potentially leading to important clinical consequences [10]. On the other hand, isavuconazole has a more predictable pharmacokinetic profile and a lower potential for DDIs compared to other azoles. Although it is a moderate inhibitor of CYP3A4, its interaction profile is generally more favorable, making it a safer option in patients receiving multiple concomitant medications. Nonetheless, caution is still required when isavuconazole is co-administered with strong CYP3A4 inhibitors or inducers, as these can affect its plasma levels. Hence, TDM is not always required except in specific situations. For isavuconazole, a loading dose is also needed (372 mg po/IV q8h × 6 doses), followed by 372 mg po/IV daily [153]. This antifungal agent helps decrease susceptibility to other triazoles, as it shows fewer adverse effects and DDI than voriconazole [151]. Posaconazole is a triazole studied for IFI prophylaxis in high-risk patients and as salvage therapy in refractory fungal infections [154]. Itraconazole is used in non-invasive or chronic forms of aspergillosis or following intolerance or toxicity to other triazoles.

Echinocandins such as caspofungin, micafungin and anidulafungin, available in intravenous formulation, are not recommended as first-line agents but as salvage therapy or in combination with other drugs [155,156].

Combination therapy is a strategy to improve patient outcomes and overcome antifungal resistance, which is often caused by mutations in the target site of triazoles (CYP51A). These mutations can occur due to prolonged therapy or in regions where azoles are used on agricultural products [154,155]. In the presence of azole-resistant infection, lipid AmB with azole or echinocandin have been proposed [156].

In the treatment of aspergillosis in immunocompromised patients, we should consider the different antifungal treatment approaches, evaluate the efficacy of trials of specific drugs in this particular setting, and consider the pharmacological aspect of the available drugs to minimize pharmacological exposure and limit adverse effects [157].

## 7. Antifungal Resistance

### 7.1. Amphotericin B

Amphotericin B exerts its antifungal activity by binding to sterols, particularly ergosterol, a key component of the fungal cell membrane. This interaction disrupts membrane integrity through the formation of transmembrane pores, leading to increased permeability and the consequent leakage of intracellular ions and cytoplasmic contents. This membrane dysfunction ultimately results in fungal cell death. Additionally, amphotericin B may induce oxidative stress by generating reactive oxygen species (ROS), further contributing to cellular damage [158,159]. Generally, resistance to amphotericin B is quite rare [160].

Among non-*fumigatus Aspergillus* species, resistance to amphotericin B is mainly observed in *A. terreus* known to exhibit intrinsic resistance, although the underlying molecular mechanisms remain unclear and *A. flavus*. It has been hypothesized that the increased use of amphotericin B, partly driven by the emergence of azole-resistant *A. fumigatus*, may have contributed to the selection and rise of amphotericin B resistance isolates in recent years [158].

### 7.2. Azoles

Since the late 2000s, azole resistance in *A. fumigatus* has emerged as a serious global concern due to increasing reports of therapeutic failure. Although resistance can develop during antifungal treatment, particularly in patients receiving long-term therapy, environmental exposure to agricultural azole fungicides is considered the primary driver of resistance selection [149,161,162,163].

Azoles act by inhibiting lanosterol 14α-demethylase, an enzyme encoded by the cyp51A gene, which is essential for ergosterol biosynthesis. Inhibition leads to the accumulation of toxic 14α-methyl sterols, compromising membrane structure and function, and ultimately impairing fungal growth [149,161,162].

Mutations in cyp51A—especially nonsynonymous point mutations—are strongly associated with azole resistance. These mutations can reduce drug binding or uptake, conferring variable resistance profiles [149,161,162].

Due to the natural process of asexual sporulation, *A. fumigatus* populations can harbor diverse genetic variants, including both azole-susceptible and -resistant strains within the same host. This genetic heterogeneity complicates treatment and contributes to the persistence of resistant clones [149,161,162].

### 7.3. Echinocandins

Echinocandins inhibit β-(1,3)-D-glucan synthase, an enzyme involved in fungal cell wall synthesis. Against *Aspergillus* spp., echinocandins exhibit only fungistatic activity, which limits their utility as monotherapy. They are primarily employed in combination with azoles or polyenes to achieve a synergistic antifungal effect. Resistance to echinocandins among *Aspergillus* spp. remains rare but should still be monitored, especially in cases of prolonged exposure or combination therapy [158,164].

## 8. Conclusions

In recent years, our understanding of pulmonary aspergillosis in immunocompromised patients has significantly evolved. This narrative review highlights the emergence of new host populations at risk—including non-neutropenic ICU patients and individuals with viral infections such as COVID-19 and influenza—who present with atypical forms of invasive aspergillosis. Diagnostic approaches have progressed beyond traditional culture-based methods, combining different tools such as galactomannan, 1,3-β-D-glucan, PCR, and lateral flow assays. Nevertheless, significant diagnostic delays persist, especially in non-classical hosts, due to limitations in specificity and accessibility of these tools in critically ill patients.

Currently, treatment options for invasive fungal diseases (IFDs) remain limited, and resistance to existing antifungal agents is increasingly common among pathogenic fungi. In particular, the emergence of azole-resistant *A. fumigatus* represents a significant public health concern in Europe [158]. Globally, it is estimated that up to 19% of *A. fumigatus* infections are resistant to azole therapy [165].

The growing threat of azole resistance in *A. fumigatus* is mirrored by a broader rise in antifungal resistance across non-*fumigatus Aspergillus* species and cryptic isolates that can also be intrinsically resistant to antifungal therapies. These developments highlight the urgent need for advanced molecular diagnostics and routine susceptibility testing to ensure timely and appropriate therapeutic decisions.

A new promising drug, Olorofim, has been developed. It is the first antifungal agent in the orotomide class and has demonstrated activity against fungi resistant to approved treatments. It acts by disrupting fungal pyrimidine biosynthesis, ultimately causing cell death. Preliminary data showed the efficacy and the safety of Olorofim for the treatment of IFD in patients with few or no treatment options [166]. Advances in antifungal pharmacology includes also newer triazoles and combination therapies that are reshaping treatment strategies but require further validation in high-risk populations.

Ultimately, improving outcomes for immunocompromised patients with aspergillosis demands a multidisciplinary approach: integrating early risk stratification, timely use of molecular diagnostics, and personalized antifungal treatment. Future research should prioritize prospective studies in emerging risk groups, standardized diagnostic criteria for ICU populations, and strategies to mitigate antifungal resistance globally.

## Figures and Tables

**Figure 1 jof-11-00617-f001:**
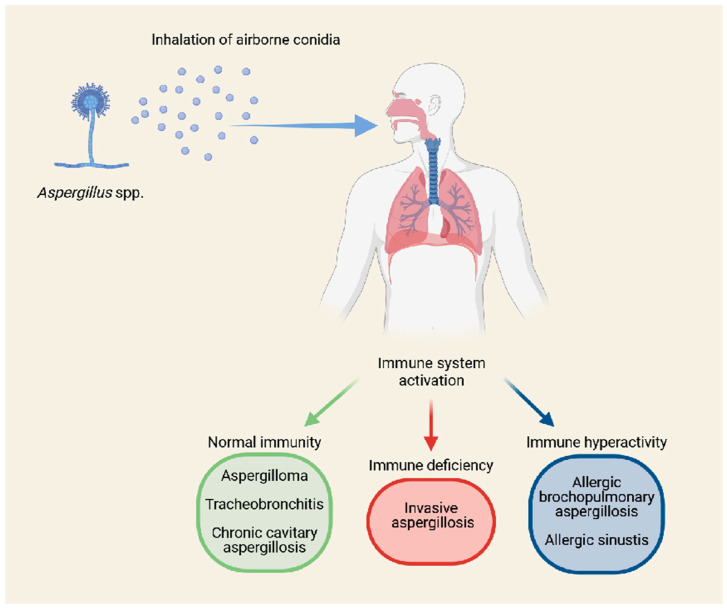
Spectrum of disease due to interaction between *Aspergillus* and the human host. The inhalation of *Aspergillus* spp. airborne conidia can cause different categories of human disease depending on the immunological status of the infected host. Among immunocompetent individuals, diseases such as aspergilloma, tracheobronchitis, or chronic cavitary pulmonary aspergillosis (CCPA) are typically caused by *A. fumigatus*, but other species such as *A. niger*, *A. flavus*, and *A. terreus* may also be involved. In immunocompromised patients, IA is predominantly associated with *A. fumigatus*, but emerging pathogens like *A. terreus*, *A. nidulans*, *A. flavus*, and cryptic species in the *A. fumigatus* complex (e.g., *A. lentulus*, *A. udagawae*) have increasingly been reported, often associated with intrinsic antifungal resistance. Conversely, patients with immune system hyperactivation may develop allergic manifestations such as severe asthma with fungal sensitization (SAFS), allergic bronchopulmonary aspergillosis (ABPA), and allergic fungal rhinosinusitis (AFRS), mostly related to *A. fumigatus* and occasionally *A. niger*.

**Figure 2 jof-11-00617-f002:**
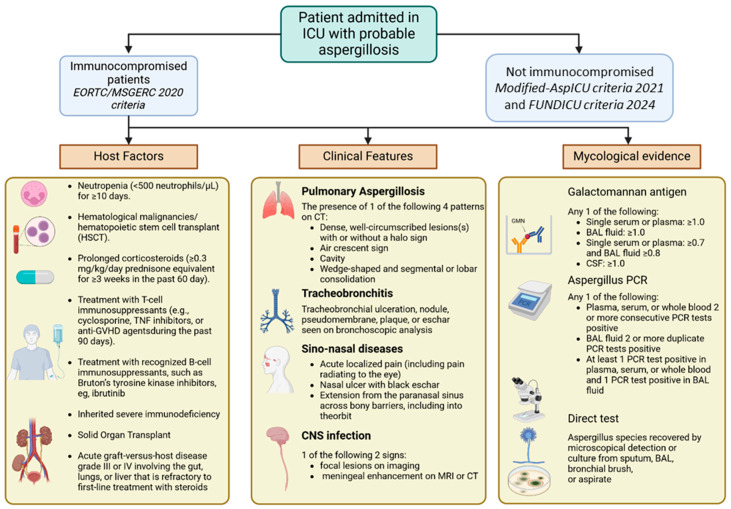
Criteria for probable aspergillosis diagnosis in patients admitted in ICU. Various criteria have been proposed for diagnosing probable aspergillosis in different patient populations. Modified AspICU criteria 2021 and FUNDICU criteria 2024 are applied to non-immunocompromised individuals. The EORTC/MSGERC criteria are used for immunocompromised patients and rely on a combination of host factors, clinical and radiological evidence, and microbiological and histopathological findings to achieve a rapid and accurate diagnosis. For a proven diagnosis of aspergillosis, identification of *Aspergillus* is required through microscopic analysis of sterile material, histopathological or cytopathological examination, or direct microscopic examination of a specimen obtained via needle aspiration or biopsy. Possible aspergillosis requires at least 1 host factor and 1 clinical criterion without any mycological criteria. CSF: Cerebrospinal fluid.

**Figure 3 jof-11-00617-f003:**
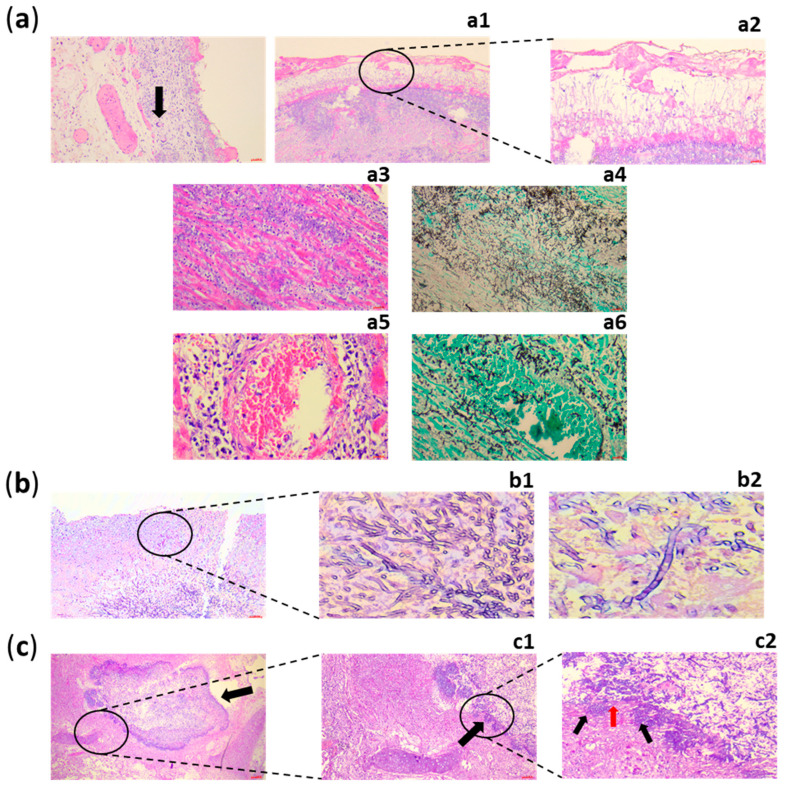
Direct microscopic diagnosis of IA in three immunocompromised patients. (**a**) Patient n. 1 HSCT recipient presenting IA and a granulomatous reaction with giant cells (black arrow) surrounding *Aspergillus* hyphae on the surface of the pleura. (**a1**) Splendor Hoeppli phenomenon (black circle) around the hyphae. This phenomenon consists of radiating eosinophilic material at the lesion’s periphery and is typical of chronic granulomatous lesions. (**a2**) Detail of (**a1**): Splendor Hoeppli phenomenon with septated hyphae and conidial heads. (**a3**) Cardiac dissemination of *Aspergillus* hyphae dissecting myocardial fibers with neutrophilic inflammation (myocarditis). (**a4**) Hyphae of *Aspergillus* highlighted by histochemical reaction of Grocott-Gomori methenamine silver (GMS). (**a5**,**a6**): (**a5**) hematogenous spread of *Aspergillus* in IA. (**a6**) Hyphae within a vascular lumen highlighted by GMS. (**b**) Patient n. 2 with AIDS presenting necrotizing pseudomembranous bronchitis with pseudomembrane of *Aspergillus* in bronchial wall eroding the epithelium (black circle; exudate of necrotic tissue, inflammatory cells, fungal hyphae and conidial heads). Figure (**b1**,**b2**) show an enlargement of figure (**b**) of the typical branching at acute angle shape (45°) of the *Aspergillus* hyphae with conidial heads. Hyphae of *Aspergillus* are branching (**b1**) and septated (**b2**). (**c**) Patient n. 3 positive to SARS-CoV-2 with necrotizing pseudomembranous bronchitis. The fungus plug (black arrow) completely occludes the airway lumen. Post-mortem artifacts are present. (**c1**) Enlargement of pseudomembrane of *Aspergillus* growing along the surface of the bronchial mucosa and eroding the epithelium (black arrow; exudate of necrotic tissue, inflammatory cells, fungal hyphae and conidial heads). (**c2**) Detail of (**c1**): fungal hyphae and conidial heads (red arrow) eroding the mucosa (black arrows). Images were taken from the hospital archive.

**Table 1 jof-11-00617-t001:** Reliability of GM test with ODI 0.5 for serum/plasma and 1.0 for BAL.

Sample/Test	Sensitivity	Specificity
Serum/plasma (proven/probable vs. no IPA)	76%	92%
Serum/plasma (proven/probable/possible vs. no IPA)	45%	91%
BAL (proven/probable vs. no IPA)	80%	95%
BAL (proven/probable/possible vs. no IPA)	49%	95%

**Table 2 jof-11-00617-t002:** Comparison of Diagnostic Criteria: EORTC/MSGERC 2020 vs. Modified AspICU 2021 vs. FUNDICU 2024.

	EORTC/MSGERC (2020)	Modified AspICU (2021)	FUNDICU (2024)
**Target population**	Severely immunocompromised (e.g., neutropenic, HSCT, hematologic malignancies)	Non-neutropenic ICU patients	All critically ill ICU patients, included non-classically immunocompromised
**Diagnostic categories**	Possible, Probable, Proven	Proven, Probable, Colonization	Possible, Probable, Proven
**Required clinical criteria**	Not central: diagnosis mostly relies on host factors, imaging and microbiological criteria	Yes: fever, respiratory symptoms, worsening oxygenation	Yes: detailed ICU-adapted criteria (e.g., sepsis, new infiltrates, secretions)
**Radiological criteria**	Yes: typical signs such as halo, air crescent, cavitation	Yes: infiltrates or new lesions consistent with infection	Yes: including HRCT or other imaging compatible with IPA
**Microbiological criteria**	Positive culture from sterile site or BAL GM index ≥ 0.5 in serum GM index ≥ 1.0 in BAL Positive PCR histopathology showing hyphae with tissue damage	BAL/tracheal culture GM in BAL (ODI ≥1.0) PCR optional	Culture GM (BAL ODI ≥ 1.0, serum ODI ≥ 0.5) BDG (>80 pg/mL) PCR on blood or BAL histopathology if available
**Included host factors**	Strict: Neutropenia (ANC < 500 cells/μL for >10 days), allogeneic HSCT, prolonged corticosteroids, T-cell immunosuppressants, inherited immunodeficiencies, solid organ transplant, recent chemotherapy, GVHD, or treatment with anti-cytokine biologics	ICU-specific: prolonged mechanical ventilation, ARDS, chronic lung disease	Broad: includes viral pneumonia, immunotherapy, ARDS, prolonged ICU stay, chronic lung disease
**Advanced diagnostic techniques (PCR, BDG, GM)**	Yes (GM, BDG, PCR, histopathology)	Partially, BAL GM included, PCR optional	Yes, required for Probable classification
**Recognition of viral co-infections**	No	Partially: influenza included, COVID-19 not systematically	Yes: COVID-19 and influenza recognized
**Validation and intended use**	Standard in trials and guidelines for immunocompromised patients	Used in ICU research, CAPA/IAPA definitions	Built by international consensus, intended for ICU clinical research and standardization

**Table 3 jof-11-00617-t003:** Non-culture test characteristics.

GM in Serum Samples	GM in BAL Samples	1,3-β-D-Glucan	Molecular Methods (PCR)	Lateral Flow Device (LFD) Assay
Best performance in neutropenic patients (ODI > 0.5 in two samples)	Useful in non-neutropenic patients (no positive serum GM)	Nonspecific marker	Detection of several *Aspergillus* spp. in immunocompromised individuals	*Aspergillus*-specific monoclonal antibody for the detection of an extracellular glycoprotein secreted by *Aspergillus* spp., during active growth
Screening test in patient at risk	Cut-off not established: 0.5 USA versus 1.0 Europe in relation to patients’ risk	High NPV, useful as screening in high-risk patients	Early diagnosis with a high NPV, high sensitivity	Serum and BAL are the samples
Serum GM present 5–8 days before clinical manifestations	Performance test depends on technique used for BAL procedure (sterile saline volume instilled during bronchoscopy, volume and type of collected BAL fluid), mold prophylaxis or therapy, risk of fungal colonization		It allows to quantify and to recognize *Aspergillus* species	Cross-reactivity with *Penicillium*
Serum GM should not be used on patients at risk but on mold active prophylaxis	Major sensitivity than serum GM assay with a high PPV		High cost and technical difficulties	Rapid and cost-effective

## Data Availability

No new data were created or analyzed in this study. Data sharing is not applicable to this article.

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
