# Peer review of "Pulmonary Aspergillosis in Immunocompromised Critically Ill Patients: Prevalence, Risk Factors, Clinical Features and Diagnosis—A Narrative Review"

_jof, 2025, doi:10.3390/jof11090617_

Round 1
Reviewer 1 Report
In the early section of the review the authors address what is known about immunity to Aspergillus and they cover aspects of innate and adaptive immune responses. However, of necessity, the review in this section is a bit limited. For example there is no mention of the role of Toll receptors, or Dectin 1/2 (page4).
In most areas Aspergillus is italicised but in some other areas of the manuscript it is not. This should be addressed for consistency.
Page 5: there is reference to "melanized yeast like forms" are these actually yeast like or conidia like?
Table 1 refers to plasma galactomannan when describing the performance of the assay in terms of sensitivity and specificity; I think most studies have conducted galactomannan testing on serum rather than plasma? Perhaps they could instead use: serum/plasma?
Page 9 line 371: note typo for respectively. Nearby, abbreviation PHW should be PWH. Line 379: within six months prior to surgery
Page 11 line 449, the sentence patients...were mostly 62 years old, this sounds unlikely that most patients were the same age, do they mean were mostly more than 60 years of age?
lines 472 and 474 the description of IAPA is duplicated
Line 490: ? typo in sentence referring to excessive type interferon gamma responses
Page 14 line 557: what does HIV stadium C3 mean?
Page 15 line 568: there cannot by "almost one host factor"
line 596: typo "oinfections"
line 651 change sensibility to sensitivity
Line 654: there is mention that piperacillin-tazobactam causes false positive serum/plasma galactomannan results. My understanding is that this problem has been resolved with current batches of Platelia galactomannan?
Line 673: what is the relevance of interstitial cystitis to the topic in this section?
line 709: ? typo. is the test helping to distinguish probable from proven IPA?
Page 20 line 7: by susceptibility do the authors mean immunity?
There is mention of voriconazole being the drug of choice for treating IPA but in international guidelines isavuconazole has the same first-line recommendation
See above.
Author Response
REVIEWER 1
Comment 1: The authors have produced a review on pulmonary aspergillosis covering the the stated topics of prevalence, risk factors and clinical features. They address both classically immunocompromised (eg, leukaemia, HSCT) patients as well as the emerging groups including patients in intensive care with or without severe predisposing respiratory viral infections (influenza, COVID-19). The review actually spends a substantial amount of time covering diagnosis, so I would suggest that "diagnosis' should be included somewhere in the title.
Response 1: We thank the Reviewer for this insightful comment. We have now added “diagnosis” in the title.
Comment 2: The conclusion section is a bit limited. It describes diagnostic tests such as galactomannan as novel whereas in fact this test, in one or more formats. has been in use for 30 years or more. There is brief reference to antifungal drug resistance, but no real coverage of this topic in the main review. The authors highlight resistance in non-fumigatus Aspergillus spp whereas the main problem with resistance is in A. fumigatus (to triazoles) There is no mention of new drugs in development like Olorofim which should be included in any up to date review if the intention is to cover antifungal therapy.
Response 2: We thank the Reviewer for the valuable and constructive comments. As suggested, we have revised the Conclusion section to better reflect the current state of the field, removing the characterization of galactomannan as a “novel” diagnostic tool. Furthermore, we have added a dedicated paragraph on antifungal drug resistance (section 7, lines 53-91), with particular attention to Aspergillus fumigatus and triazole resistance, which indeed represents the main concern. We have also included a brief mention of Olorofim, as a new antifungal agent currently in development, to ensure the review remains up to date with ongoing therapeutic advancements.
Updated text in the manuscript:
- Antifungal resistance (lines 53-91)
Amphotericin B
Amphotericin B exerts its antifungal activity by binding to sterols, particularly ergosterol, a key component of the fungal cell membrane. This interaction disrupts membrane integrity through the formation of transmembrane pores, leading to increased permeability and the consequent leakage of intracellular ions and cytoplasmic contents. This membrane dysfunction ultimately results in fungal cell death. Additionally, amphotericin B may induce oxidative stress by generating reactive oxygen species (ROS), further contributing to cellular damage [161, 162]. Generally, resistance to amphotericin B is quite rare [163].
Among non-fumigatus Aspergillus species, resistance to amphotericin B is mainly observed in A. terreus known to exhibit intrinsic resistance, although the underlying molecular mechanisms remain unclear and A. flavus. It has been hypothesized that the increased use of amphotericin B, partly driven by the emergence of azole-resistant A. fumigatus, may have contributed to the selection and rise of amphotericin B resistance isolates in recent years [161].
Azoles
Since the late 2000s, azole resistance in A. fumigatus has emerged as a serious global concern due to increasing reports of therapeutic failure. Although resistance can develop during antifungal treatment, particularly in patients receiving long-term therapy, environmental exposure to agricultural azole fungicides is considered the primary driver of resistance selection [164-167].
Azoles act by inhibiting lanosterol 14α-demethylase, an enzyme encoded by the cyp51A gene, which is essential for ergosterol biosynthesis. Inhibition leads to the accumulation of toxic 14α-methyl sterols, compromising membrane structure and function, and ultimately impairing fungal growth [164-166].
Mutations in cyp51A—especially nonsynonymous point mutations—are strongly associated with azole resistance. These mutations can reduce drug binding or uptake, conferring variable resistance profiles [164-166]. Due to the natural process of asexual sporulation, A. fumigatus populations can harbor diverse genetic variants, including both azole-susceptible and -resistant strains within the same host. This genetic heterogeneity complicates treatment and contributes to the persistence of resistant clones [164-166].
Echinocandins
Echinocandins inhibit β-(1,3)-D-glucan synthase, an enzyme involved in fungal cell wall synthesis. Against Aspergillus spp., echinocandins exhibit only fungistatic activity, which limits their utility as monotherapy. They are primarily employed in combination with azoles or polyenes to achieve a synergistic antifungal effect. Resistance to echinocandins among Aspergillus spp. remains rare but should still be monitored, especially in cases of prolonged exposure or combination therapy [161,168].
- 8. Conclusion (changes highlighted in yellow)
In recent years, our understanding of pulmonary aspergillosis in immunocompromised patients has significantly evolved. This narrative review highlights the emergence of new host populations at risk—including non-neutropenic ICU patients and individuals with viral infections such as COVID-19 and influenza—who present with atypical forms of invasive aspergillosis. Diagnostic approaches have progressed beyond traditional culture-based methods, combining different tools such as galactomannan, 1,3-β-D-glucan, PCR, and lateral flow assays. Nevertheless, significant diagnostic delays persist, especially in non-classical hosts, due to limitations in specificity and accessibility of these tools in critically ill patients. Currently, treatment options for invasive fungal diseases (IFDs) remain limited, and resistance to existing antifungal agents is increasingly common among pathogenic fungi. In particular, the emergence of azole-resistant fumigatus represents a significant public health concern in Europe [161]. Globally, it is estimated that up to 19% of A. fumigatus infections are resistant to azole therapy [169]. The growing threat of azole resistance in A. fumigatus is mirrored by a broader rise in antifungal resistance across non-fumigatus Aspergillus species and cryptic isolates that can also be intrinsically resistant to antifungal therapies. These developments highlight the urgent need for advanced molecular diagnostics and routine susceptibility testing to ensure timely and appropriate therapeutic decisions. A new promising drug, Olorofim, has been developed. It is the first antifungal agent in the orotomide class and has demonstrated activity against fungi resistant to approved treatments. It acts by disrupting fungal pyrimidine biosynthesis, ultimately causing cell death. Preliminary data showed the efficacy and the safety of Olorofim for the treatment of IFD in patients with few or no treatment options [170]. Advances in antifungal pharmacology, includes also newer triazoles and combination therapies that are reshaping treatment strategies but require further validation in high-risk populations. Ultimately, improving outcomes for immunocompromised patients with aspergillosis demands a multidisciplinary approach: integrating early risk stratification, timely use of molecular diagnostics, and personalized antifungal treatment. Future research should prioritize prospective studies in emerging risk groups, standardized diagnostic criteria for ICU populations, and strategies to mitigate antifungal resistance globally.
Comment 3: In the early section of the review the authors address what is known about immunity to Aspergillus and they cover aspects of innate and adaptive immune responses. However, of necessity, the review in this section is a bit limited. For example there is no mention of the role of Toll receptors, or Dectin 1/2 (page4).
Response 3: We acknowledge the Reviewer for this comment. We have implemented the section about immunity to Aspergillus (line 138-159, line 170-192).
Updated text in the manuscript:
Section 2, lines 138-159:
The removal of fungi from an infected host depends on the rapid migration of a sufficient number of phagocytes to the site of infection and the recognition of fungal PAMPs via PRRs (both soluble and membrane-bound). This leads to the ingestion and, ultimately, the degradation of the ingested fungal cells within the phagocytes. [20]. A key aspect of this process is the maturation of conidia, which, upon swelling, lose the thin hydrophobic RodA protein layer, a surface component of Aspergillus that masks the immunogenic components of the cell wall, such as β-glucan. This allows for increased exposure of fungal PAMPs, facilitating recognition by PRRs such as Dectin-1 and Dectin-2. Dectin-1, expressed on macrophages, neutrophils, and dendritic cells, recognizes β-glucan fragments on swollen conidia, mediating a prolonged inflammatory response and the activation of immune cells. Similarly, Dectin-2, expressed on macrophages and dendritic cells, binds fungal mannans, facilitating early recognition of conidia and the subsequent recruitment of phagocytes [21].
Section 2, lines 170-192:
Cell-mediated immunity plays a crucial role in anti-fungal immunity or aller-gy-associated tissue damage. Fungal antigens are carried to lymph nodes, where they are processed and presented by antigen-presenting cells, leading to the differentiation of naive CD4⁺ T cells into distinct T-helper subsets. In the context of invasive aspergillosis, the functional polarization of these T cells is critical in determining the outcome of infection. Th1 cells, characterized by T-bet expression and the production of interferon-gamma, promote macrophage activation [29]. A robust Th1 response is generally associated with protective immunity, and its deficiency correlates with increased susceptibility to IA. For the Th17 cells, the differentiation is driven by innate cytokines such as IL-1α, IL-1β, IL-6, and IL-23, which activate the lineage-defining transcription factor RORγT (retinoic acid receptor-related orphan receptor gamma T). These Th17 cells secrete IL-17A and IL-22, promoting neutrophil recruitment and strengthening epithelial barrier defenses through the induction of antimicrobial peptides. While Th17 responses contribute to fun-gal control in the early phases of infection, excessive or prolonged activation may lead to immunopathology and tissue damage. Th2 responses, characterized by IL-4, IL-5, and IL-13 production, are considered detrimental in IA, as they interfere with protective Th1 activity and may impair antifungal effector mechanisms. Regulatory T cells (Tregs), defined by Foxp3 expression and the secretion of IL-10 and TGF-β, act to restrain excessive immune activation and limit tissue damage. However, in the setting of IA, an overabundance or overactivity of Tregs may suppress effective Th1 and Th17 responses, contributing to fungal persistence. These observations highlight the necessity of a balanced adaptive immune response in IA—one that ensures pathogen clearance while preventing immune-mediated injury [30,31].
Comment 4: In most areas Aspergillus is italicised but in some other areas of the manuscript it is not. This should be addressed for consistency.
Response 4: We thank the Reviewer for noticing this error. It has been corrected as suggested.
Comment 5: Page 5: there is reference to "melanized yeast like forms" are these actually yeast like or conidia like?
Response 5: We thank the Reviewer for the comment. Melanized yeast-like forms are atypical fungal structures that can develop in response to stress conditions such as antifungal treatments, which cause the organisms to assume rounded and melanized shapes. However, melanin is a pigment primarily found in the conidia. This definition “melanized yeast like forms” is reported in the 2008 EORTC/MSGERC definitions of invasive fungal infection for proven infection.
Comment 6: Table 1 refers to plasma galactomannan when describing the performance of the assay in terms of sensitivity and specificity; I think most studies have conducted galactomannan testing on serum rather than plasma? Perhaps they could instead use: serum/plasma?
Response 6: We thank the Reviewer for the comment. We modified the table 1, using serum/plasma, as suggested.
Comment 7: Page 9 line 371: note typo for respectively. Nearby, abbreviation PHW should be PWH. Line 379: within six months prior to surgery
Response 7: Thank you for noticing these errors. They have been corrected as suggested.
Comment 8: Page 11 line 449, the sentence patients...were mostly 62 years old, this sounds unlikely that most patients were the same age, do they mean were mostly more than 60 years of age?
Response 8: We thank the Reviewer for the comment. We have revised and corrected the sentence as follows “…. A Spanish study found the median age to be 62 years (range 48–74)”. Section 4, lines 476-477.
Comment 9: lines 472 and 474 the description of IAPA is duplicated
Response 9: We thank the Reviewer for noticing the error. It has been corrected.
Comment 10: Line 490: ? typo in sentence referring to excessive type interferon gamma responses
Response 10: We thank the Reviewer for the comment. We removed the word “type” from the sentence.
Comment 11: Page 14 line 557: what does HIV stadium C3 mean?
Response 11: We thank the Reviewer for the comment. According to the Center for Disease and Control (CDC) HIV stage CDC C3 is the final stage of HIV corresponding to AIDS (acquired immunodeficiency syndrome). We have modified the sentence to ensure the intended meaning is more accurately conveyed.
Updated text in the manuscript:
“Patient n. 2 with AIDS” line 581.
Comment 12: Page 15 line 568: there cannot by "almost one host factor"
Response 12: We thank the Reviewer for pointing out this imprecise expression. We have revised the sentence to improve clarity. The phrase “almost one host factor” has been corrected to “at least one host factor”. Line 592.
Comment 13: line 596: typo "oinfections"
Response 13: We thank the Reviewer for noticing this error. It has been corrected in “coinfections”. Line 620.
Comment 14: line 651 change sensibility to sensitivity
Response 14: We agree with this suggestion. It has been changed. Line 675.
Comment 15: Line 654: there is mention that piperacillin-tazobactam causes false positive serum/plasma galactomannan results. My understanding is that this problem has been resolved with current batches of Platelia galactomannan?
Response 15: We fully agree with the Reviewer’s observation. The issue of false-positive galactomannan results related to piperacillin-tazobactam has been significantly mitigated in recent years due to improved manufacturing processes and the evolution of the Platelia™ Aspergillus EIA assay.
Updated text in the manuscript:
Section 5, line 677.
False negatives are possible in patients treated with antifungal therapy. At the same time, false positives are feasible in patients with another fungal infection.
Comment 16: Line 673: what is the relevance of interstitial cystitis to the topic in this section?
Response 16: We appreciate the Reviewer’s observation. The sentence referring to interstitial cystitis has been removed from the manuscript, as it was not directly relevant.
Comment 17: line 709: ? typo. is the test helping to distinguish probable from proven IPA?
Response 17: We thank the Reviewer for pointing this out. We have clarified the sentence to better reflect the intended meaning. The purpose of TAFC detection is not to differentiate probable from proven IPA, but rather to support the diagnostic classification of IPA, particularly by increasing the likelihood of identifying probable IPA in the early stages. The sentence has been revised accordingly for clarity.
Updated text in the manuscript:
Section 5, lines 729-734.
Another biomarker is triacetyl-fusarinine C (TAFC), a fungal molecule produced by some molds. One of these is A. fumigatus. It is a secreted siderophore that can be used as a biomarker. In fact, its detection in bodily fluids (e.g., serum, BAL or urine) may aid the di-agnosis of IA and represents a promising tool to support the early identification of IPA in high-risk patients; if it has been combined with GM-BAL, the sensitivity and specificity of GM increased
Comment 18: Page 20 line 7: by susceptibility do the authors mean immunity?
Response 18: We acknowledge the Reviewer’s observation. We revised the sentence.
Updated text in the manuscript:
Section 6, lines 5-7.
The development of novel chemotherapy agents and immunosuppressive drugs has im-proved outcomes but also increased susceptibility to IFIs [150].
Comment 19: There is mention of voriconazole being the drug of choice for treating IPA but in international guidelines isavuconazole has the same first-line recommendation
Response 19: We acknowledge the reviewer for the comment. As suggested, we have revised the text to reflect that both voriconazole and isavuconazole are recommended as first-line treatments for invasive pulmonary aspergillosis, in accordance with international guidelines.
Updated text in the manuscript:
Section 6, lines 28-29.
Voriconazole and isavuconazole are available in oral (po) and intravenous (IV) formulations. They represent the primary regiments in IA therapy.

Reviewer 2 Report
This review manuscript provides us a comprehensive description on management of pulmonary aspergillosis in immunocompromised critically patients. The prevalence, risk factors and clinical features are well discussed. Here, a few comments as follows for your further consideration.
- In the part of immunocompromised patients, I would like to add COPD patients and cirrhosis patients who have been treated as high risk population worldwide due to its high morbidity and mortality.
- In the part of 6 regarding antifungal prophylaxis and pre-emptive treatment versus antifungal therapy, I feel not too much has been talked compared with the part of aspergillus related epidemiology and diagnosis. I would prefer input the topic of antifungal drug TDM and DDI (drug drug interaction), which play important roles in the management of IFDs in our clinical practice.
- As a whole picture of this review, I would like to talk something about new antifungal drugs which is coming soon.
- There are two minor points to be checked: Aspergillus spp. may be changed to spp. (italic font); In reference 16: the year of publication was not found.
Author Response
REVIEWER 2
Comment 1: In the part of immunocompromised patients, I would like to add COPD patients and cirrhosis patients who have been treated as high risk population worldwide due to its high morbidity and mortality.
Response 1: We thank the Reviewer for this helpful comment. While we fully agree that patients with COPD and cirrhosis (i.e., with major chronic conditions) represent populations at increased risk—particularly due to their high morbidity and mortality—they fall, within the scope of our review, under the broader category of critically ill patients rather than classically immunocompromised hosts. This distinction aligns with the specific focus of our manuscript on immunocompromised individuals as defined by underlying immune suppression rather than chronic critical illness.
Comment 2: In the part of 6 regarding antifungal prophylaxis and pre-emptive treatment versus antifungal therapy, I feel not too much has been talked compared with the part of aspergillus related epidemiology and diagnosis. I would prefer input the topic of antifungal drug TDM and DDI (drug drug interaction), which play important roles in the management of IFDs in our clinical practice.
Response 2: We thank the Reviewer for this insightful suggestion. We have expanded Section 6 by adding content on TDM and DDI, acknowledging their critical role in the management of invasive fungal diseases in clinical practice.
Updated text in the manuscript:
Section 6, lines 28-44
Voriconazole and isavuconazole are available in oral (po) and intravenous (IV) formulations. They represent the primary regiments in IA therapy. A loading dose is necessary for voriconazole: 6 mg/kg IV every 12 hours for two doses, followed by 4 mg/kg every 12 hours. Therapeutic drug monitoring (TDM) is recommended due to its complex metabo-lism and numerous drug–drug interactions (DDIs), primarily due to its inhibition of CYP450 enzymes. These interactions can significantly alter the plasma concentrations of both voriconazole and co-administered drugs, potentially leading to important clinical consequences [152]. On the other hand, isavuconazole has a more predictable pharmaco-kinetic profile and a lower potential for DDIs compared to other azoles. Although it is a moderate inhibitor of CYP3A4, its interaction profile is generally more favorable, making it a safer option in patients receiving multiple concomitant medications. Nonetheless, caution is still required when isavuconazole is co-administered with strong CYP3A4 inhibitors or inducers, as these can affect its plasma levels. Hence, TDM is not always re-quired except in specific situations. For isavuconazole is also needed a loading dose (372 mg po/IV q8h x 6 doses) followed by 372 mg po/IV daily [156]. This antifungal agent helps decrease susceptibility to other triazoles, as it shows fewer adverse effects and DDI than voriconazole [154].
Comment 3: As a whole picture of this review, I would like to talk something about new antifungal drugs which is coming soon.
Response 3: We acknowledge the reviewer for the comment. We have included in section 8 (conclusion) a brief mention of Olorofim, as a new antifungal agent currently in development, to ensure the review remains up to date with ongoing therapeutic advancements.
Updated text in the manuscript:
Section 8, lines 120-124 (changes highlighted in yellow)
These developments highlight the urgent need for advanced molecular diagnostics and routine susceptibility testing to ensure timely and appropriate therapeutic decisions. A new promising drug, Olorofim, has been developed. It is the first antifungal agent in the orotomide class and has demonstrated activity against fungi resistant to approved treatments. It acts by disrupting fungal pyrimidine biosynthesis, ultimately causing cell death. Preliminary data showed the efficacy and the safety of Olorofim for the treatment of IFD in patients with few or no treatment options [170].
Comment 4: There are two minor points to be checked: Aspergillus spp. may be changed to spp. (italic font); In reference 16: the year of publication was not found.
Response 4: We thank the reviewer for noticing these errors. They have been corrected.
